# Influence of Surface Chemical and Topographical Properties on Morphology, Wettability and Surface Coverage of Inkjet-Printed Graphene-Based Materials

**DOI:** 10.3390/mi15060681

**Published:** 2024-05-22

**Authors:** Iulia Salaoru, Dave Morris, Ecaterina Ware, Krishna Nama Manjunatha

**Affiliations:** 1Emerging Technologies Research Centre, School of Engineering and Sustainable Development, Faculty of Computing, Engineering and Media, De Montfort University, Leicester LE1 9BH, UK; davewfmorris@gmail.com (D.M.); krishna.namamanjunatha@dmu.ac.uk (K.N.M.); 2Department of Materials, Faculty of Engineering, Imperial College London, South Kensington Campus, London SW7 2AZ, UK; e.ware@imperial.ac.uk

**Keywords:** inkjet printing, graphene ink, graphene oxide ink, morphology, wettability, surface coverage

## Abstract

The inkjet printing of water-based graphene and graphene oxide inks on five substrates, two rigid (silicon and glass) and three flexible (cellulose, indium tin oxide-coated polyethylene terephthalate (ITO-PET) and ceramic coated paper (PEL paper)), is reported in this work. The physical properties of the inks, the chemical/topographical properties of selected substrates, and the inkjet printing (IJP) of the graphene-based materials, including the optimisation of the printing parameters together with the morphological characterisation of the printed layers, are investigated and described in this article. Furthermore, the impact of both the chemical and topographical properties of the substrates and the physical properties of graphene-based inks on the morphology, wettability and surface coverage of the inkjet-printed graphene patterns is studied and discussed in detail.

## 1. Introduction

Recent developments in the deposition of functional materials and the fabrication of electronic devices have focused on low-cost, simple, eco-friendly and energy-saving processes. Printed electronics are emerging as a novel and sustainable platform for the fabrication of electronic devices, and Inkjet Printing (IJP) fits perfectly within this framework. The main benefits of inkjet printing manufacturing include the following: a reduction in wasted material, cost-effectiveness, compatibility with a wide range of substrates, digital and additive deposition, mask-less depositions, as well as small to large area deposition [1]. Due to these benefits, inkjet printing can be defined as a sustainable and environmentally friendly approach when compared to the current manufacturing technology, and currently it is intensively explored for the deposition of functional materials and electronic devices, with a huge potential to replace totally or partially traditional manufacturing technologies. The impact of the 3D printer technologies in the landscape of electronics has been presented in comprehensive review papers [2,3,4].

Inkjet printing (IJP) is an additive manufacturing technique that works through generating small ink droplets and propelling those droplets onto a substrate [5,6,7,8]. The three main components of inkjet printing technology consist of the following: ink, printhead and the substrate. All these components play a crucial role in the printing process. This process can be broken down into four steps, as illustrated in Table 1.

The *droplet formation* and the *droplet ejection* are the first two steps in the printing process. Firstly, these steps are influenced by the properties of the ink such as viscosity and surface tension.

Secondly, the appropriate printhead (thermal or piezoelectric) and the suitable printer head driving voltage waveform parameters (the pulse width and the amplitude of the pulse) are essential for an accurate, reliable and replicable droplet formation and ejection. Furthermore, the ink characteristics and the properties of the substrate have a strong impact on the third step: *Drops on the substrate and wetting*. In this step, it is important to achieve a good balance between the surface tension of the ink and the surface energy (chemical energy) of the substrate to achieve high-quality printed patterns. The *film formation* is the last step in the deposition. In the last two steps, wettability and adhesion are the key physical phenomena involved. Here, the topographical properties of the substrate, such as surface roughness, significantly affect the performance and quality of these layers, which is attributed to delamination as a consequence of poor adhesion, variations in the film thickness, changes in the film’s structure such as grain size and/or orientation of the printed material.

It is important to highlight that the interaction between ink and substrate and the physical properties of the ink play a key role in the formation of a continuous film when an inkjet printing method is used. Kin et al. [9] investigated the interface adhesion properties/mechanism of silver nanopaste on silicon substrate. The authors used a screen-printing technique to deposit silver nanopaste film and found that the adhesion can be enhanced when two conditions are fulfilled: firstly, the enhancement of surface roughness via a sintering temperature and secondly, including the right amount of the organic solvent at the interface. The printed silver films were subjected to a thermal treatment at 70 °C for 10 min. Furthermore, Krainer et al. [10] investigated the effect of ink properties such as surface tension and viscosity on inkjet printed picolitre dots. In their work, four different wood-free papers were used as a substrate. The authors found that the viscosity has a strong impact on drop spreading, penetration and liquid surface coverage for all types of tested substrates. The surface tension of ink has a significant influence on drop spreading/liquid penetration only when a hydrophobised substrate has been used. The roughness of the substrates and its influence on the electrical properties of Poly(3,4-ethylenedioxythiophene) polystyrene sulfonate (PEDOT:PSS) thin films printed on the paper substrate have been investigated by Morais et al. [11]. Bond paper, vegetal paper and sheets of Polyethylene Terephthalate (PET) were used as a substrate in this work. The conclusion was that substrate roughness has a strong impact on the PEDOT:PSS conductivity. The cohesion and adhesion between the silver ink and fluorocarbon-coated surface have been investigated by Lee and Cho [12]. The authors demonstrated that the surface energy alone is not sufficient to characterise the adhesion and cohesion and that the interfacial relationship between ink and the substrate should be instead investigated. Additionally, the impact of UV/O_3_ treatment on the surface of the substate on the printing quality is investigated in [12]. Kim et al. [13] performed a study on identifying the appropriate surface treatment agent composition to improve the direct to garment inkjet printing properties (colour strength, colour fastness, pigment fixation) on cotton. The authors concluded that the optimal composition should be accomplished to reduce the deviation in the total colour strength and ink smearing. Additionally, Sanatgar et al. [14] investigated the adhesion properties of polymers and nanocomposites on textiles. In their work, a 3D printing technique called fused deposition modelling (FDM) was used to print the patterns on the synthetic substrates. This work demonstrated that in the case of the FDM, the most important parameters that affect the adhesion properties are process parameters such as the following: the extruder temperature and printing speed. Yet, bendable circuits were inkjet printed by Sun et al. [15]. The authors investigated the ink (silver amine ion aqueous solution) and substrate (liquid polydimethylsiloxane) and they observed that a small amount of surfactant helps to improve the wetting behaviour of the ink on the substrate.

To date, there has been a significant and continuous effort on the development of graphene-based inks, with some results presented in [16,17,18,19,20,21,22,23].

This study investigates the influence of both the properties of the substrate and the properties of the ink and proves their respective impacts on the quality/surface coverage with a uniform and continuous pattern of graphene and graphene oxide inkjet printed multilayers. The topography (surface roughness) and chemistry (surface-free energy) of the five above-mentioned substrates were investigated by employing atomic force microscopy (AFM) and an Attention Theta Optical Tensiometer. The substrates selected did not receive any treatment before or after printing. Additionally, the physical properties (viscosity, surface tension and pH) of graphene and graphene oxide inks were evaluated in this work. The inkjet printing of the graphene-based inks, including the optimisation of the printing parameters, i.e., amplitude and pulse width, together with the morphological characterisation of printed layers, are described in this article. In a nutshell, the inkjet printing of graphene-based inks has a huge potential for the development of unique and practical electronic components.

## 2. Materials and Methods

Firstly, the physical properties of both the reduced graphene oxide (rGO) and graphene (G) inks were evaluated. The rGO is stabilised with poly(sodium 4-styrenesulfonate) 10 mg/mL dispersion in H_2_O and was purchased from Sigma Aldrich. The graphene ink is a water-based solution with 0.1wt% total solids content and was purchased from Cambridge Graphene Ltd. The first step was to check the suitability of both inks with the inkjet printing process. For the validation of the inks, the main properties such as surface tension, viscosity and pH were tested. The viscosity was evaluated via a Brookfield DV2T viscometer, using a small sample adapter with a sample volume of 16 mL and SSA 18 spindle. The surface tension of the inks was measured using a tensiometer (Torsion Balance), and 30 mL of ink was examined with a platinum ring. Yet, the pH of 30 mL ink was tested with a Jenway 3520-pH meter.

Secondly, the wettability behaviour of both inks on rigid as well as on flexible substrates was evaluated by measuring the ink/substrate contact angle using the sessile drop analysis method (Attention Theta Optical Tensiometer, Biolin Scientific, Göteborgs, Sweden). Additionally, the viscometer, pH meter, torsion balance and the optical tensiometer were calibrated before the tests were performed. Furthermore, the surface-free energy of all the selected substrates were calculated based on data from Sessile Drop experiment. Additionally, the surface roughness of both rigid and flexible substrates was evaluated via Non-contact Atomic Force Microscopy (NC-AFM). A thermal inkjet printer, ThallosJet Flatbed A3, was used to deposit both graphene and graphene oxide multi-layers patterns.

Printing graphene materials-based ink: Both graphene and graphene oxide inks were sonicated in an ultrasonic bath for several hours before printing, to ensure a good homogeneity. The printer cartridge was then filled with ink and a different number of the layers were printed. The quality of the inkjet-printed G and rGO multi-layers were assessed by using an optical microscope (LAOPHOT-2) fitted with a Nikon camera DS-Fi1 and a high-resolution field emission gun scanning electron microscope (FEG SEM) LEO Gemini 1525.

## 3. Results and Discussion

The ink, printhead and substrate(s) are the three essential elements that will be individually evaluated to ensure that all are suitable with the requirements of the IJP hardware. In order to comprehend the inkjet printing process, as illustrated in Figure 1b, the properties of ink, both the chemical and topographical properties of the substrates and the printhead waveform parameters will be compiled first.

The physical properties of both inks: surface tension, viscosity and pH were measured and are shown in Table 2.

Secondly, five substrates, two rigid (silicon and glass) and three flexible (cellulose, indium tin oxide-coated polyethylene terephthalate (ITO/PET) and PEL paper), were evaluated by measuring the surface roughness, surface-free energy and wettability.

It is important to highlight that flexible substrates were selected as today they play a crucial role in the field of flexible/paper electronics. On the other hand, silicon and glass are still used in the field of electronics, so in a nutshell, this study will provide a comprehensive study covering both types of substrates.

Figure 2 illustrates the AFM topography images for the five selected substrates. The morphologies of the chosen substrates are compared at the same scale in the x and *y*-axis without changing any measurement parameters while performing AFM topography scanning. Both the silicon and glass show a very low RMS roughness. However, the highest RMS roughness is displayed by the PEL paper. The high surface roughness in the PEL paper is possibly due to the ceramic (inorganic) coated layer applied on the top of the paper. This coated layer has the role of promoting chemical interactions that enhance sintering. Additionally, the inorganic coated paper can be heated to 150 °C (low temperature sintering) with minimal discolouration. The values of the average RMS roughness determined from AFM images of the five tested substrates are comparatively summarised in Figure 3b. Additionally, it was identified during analysis (Figure 2e) that cellulose has a number of scratch marks on its surface. It should be highlighted that the rest of the tested substrates do not display any detectable scratches or damage to their surfaces.

Furthermore, the surface-free energy of the selected substrates was tested via a sessile drop analysis method (Optical Tensiometer, Biolin Scientific, Göteborgs, Sweden), with the results presented in Figure 3a.

As can be observed from Figure 3a, ITO/PET has the lowest surface energy of 32 mN/m when compared with the rest of the substrates. On the other hand, the substrate with the highest surface-free energy 54.4 mN/m is PEL paper.

Furthermore, wettability and ink–substrate interaction have been evaluated via contact angle measurement. A low contact angle indicates that the ink spreads out more on the substrate surface and hence has a positive impact on ink spreading, leading to great uniformity and coverage.

The contact angle of a single droplet of ink with each substrate was recorded to evaluate the wetting behaviour. The values of the contact angle, collected at 60 s after the first contact, was established between inks and the substrates are presented in Figure 4. The lowest contact angle and hence a good level of wettability has been observed for both the graphene and GO inks on PEL paper.

Finally, the printing parameters such as pulse amplitude and pulse width were evaluated in order to achieve a viable print. A ThallosJet Flatbed thermal printer was used in this work. The pulse amplitude was varied from 11 V to 13 V and pulse widths were varied from 1.5 μs to 2.5 μs (see Table 3); these were tested to identify the most viable and appropriate pulse amplitude and pulse width combination for the high-quality printing of G and rGo inks. The applied voltage range was chosen between 11 V and 13 V with increments of 0.5 V. Each voltage level has a range of options for the pulse width duration periods; they were set to 1.5, 2.0 or 2.5 μs. In the case of graphene ink, 1 and 5 printed layers with the same shape and dimension (3 cm × 3 cm) were printed and the quality of the printed patterns was visually evaluated (see Figure 5a). For graphene ink, a voltage of 11 V was too low to fire the droplets of ink out of the cartridge. Similarly, a voltage of 13 V was too high as a large amount of ink was ejected and resulted in large droplets on the substrate which then spread out into one-another, creating a blotchy and non-uniform pattern. The suitable voltage that produced a deposition with greater uniformity on the PEL substrate was 11.5 V with a pulse width of 2 µs. This was selected as an appropriate and optimised pulse voltage/width setting for future printing. In contrast, for the rGO, the 11 V and 12 V appeared too low to eject the ink out of nozzles. The higher voltages were suitable for ejecting GO ink, and ink was seen ejected at 12.5 and 13 V, respectively (Figure 5b).

For a comprehensive understanding of the impact of both the properties of the substrate and ink on the wettability, the surface-free energy, roughness and surface tension of the ink are analysed further, as discussed in Figure 6.

*Surface-free energy/Surface tension*: the interactions between the ink and substrate play a key role on the wettability and hence on the printing quality. It is well known that for good wetting between the ink and substrate, the surface energy of the substrate should exceed the surface tension of the ink (by 10–15 mN/m) [24]. For the rGO ink, the substrates that follow this rule are the cellulose and the PEL paper. Conversely, for graphene ink, none of the substrates obey this rule. Furthermore, the surface energy is defined by the chemistry of the surface and a good wetting behaviour implies a surface with high energy. In the current study, the cellulose and the PEL paper are the substrates with the highest surface energy. On the other hand, during the printing process, and more particularly at Step-3—*Drops on the substrate* (see Table 1)—two competing processes are involved. Firstly, the spreading of the ink on the substrate and secondly the penetration of the ink into the substrate. At this point in the process, the topographical properties (roughness) of the substrate play a crucial role, more specifically in the ink penetration into the substrate’s pores via diffusion and capillarity penetration processes. The ink penetration into pores of the substrates that have a high RMS value of roughness promotes the formation of the mechanical interlocking interface that determines a strong adhesion between the substrate and printed pattern. Based on AFM results, presented in Figure 3b, the PEL paper has the highest roughness out of all five tested substrates. Another key observation is that the wettability can be defined and characterised by the contact angle [25]. The arriving droplets on the substrate should undergo two steps: firstly, a liquid–liquid interface evolution, and secondly, droplets coalescence. Due to the small contact angle, these steps are fast and uniform, and void free-coating printing patterns can be achieved. On the other hand, it is more difficult to obtain a high-quality pattern when the contact angle is high. Both the liquid–liquid interface/droplets coalescence is relatively slow and usually needs other factors such as temperature to sustain the coalescence process. *Wenzel* [26] demonstrated the impact of the roughness on the wettability; the wettability will be enhanced by a substrate that has high surface roughness. In this work, the contact angle data (18.8 degrees for rGO and 25.6 degrees for G ink) indicate an excellent wettability of both inks to the PEL paper. As in the current study, the substrates with the highest surface energy are both the PEL paper and cellulose. It should be highlighted that what makes the difference here and fully defines the wettability is the roughness of the substrates. In light of this, the lowest contact angle for both of the inks (Figure 4) was observed when PEL was used as a substrate. So, we can conclude that in order to identify the suitable substrate, it is imperative that both the surface energy as well as the roughness should be evaluated.

The morphology and the surface coverage of five printed layers with the same shape and dimension (3 cm × 3 cm) printed on all five selected substrates were characterised using an optical microscope and SEM. In Figure 7, optical microscope images of all printed patterns are shown. It can be clearly seen that a better coverage was achieved when the GO ink was printed, and these results are supported by the good relationship between surface-free energy and surface tension. GO printed on glass, silicon, cellulose and ITO/PET has a neck shape spreading behaviour that is defined by the liquid–liquid interface evolution during droplets coalescence.

The graphene ink printed on glass, cellulose and ITO/PET has round shaped edges and does not display any sign of neck shape spreading. However, independently from the ink used, an excellent surface coverage was achieved when PEL paper was used as the substrate. Another key observation is that the roughness of the substrate has a huge impact on the flatness of the printed patterns. As can be clearly seen from Figure 7, both the graphene oxide and graphene pattens printed on PEL paper exhibit the most significant uneven topography when compared with the rest of the used substrates.

The SEM images, presented in Figure 8, highlight a uniform and continuous pattern with interconnected flakes. Such an interconnection is most preferred for the fabrication of electronic devices as resistive losses are minimised, which facilitates the flow of current across the film more uniformly.

## 4. Conclusions

In the current work, we have investigated the impact of both the chemical and topographical properties of substrates and the physical properties of graphene-based inks on the morphology, wettability and surface coverage of the inkjet-printed patterns. The topology (roughness) of five different substrates has been measured using AFM and out of all evaluated substrates, the PEL is the substrate with the highest roughness. Furthermore, the surface-free energy of selected substrates was evaluated with the ITO/PET having the smallest surface energy and the PEL paper having the highest. Additionally, the properties (pH, viscosity and surface tension) of the graphene and graphene oxide inks were tested. A thermal inkjet printer, ThallosJet Flatbed, was used to print graphene patterns; the quality of the printed patterns and surface coverage were then evaluated via optical microscope and SEM analysis. The experimental results demonstrated that a good surface coverage with a uniform and continuous pattern can be achieved in an inkjet printing process by using a porous or high roughness substrate, complimenting the surface-free energy of the substrate with the surface tension of the selected ink. We envisage that this research will provide a good source of knowledge in the field of inkjet printing and more specifically on the impact of the topography and chemistry of a substrate on the quality, continuity, and surface coverage of the inkjet printed patterns.

## Figures and Tables

**Figure 1 micromachines-15-00681-f001:**
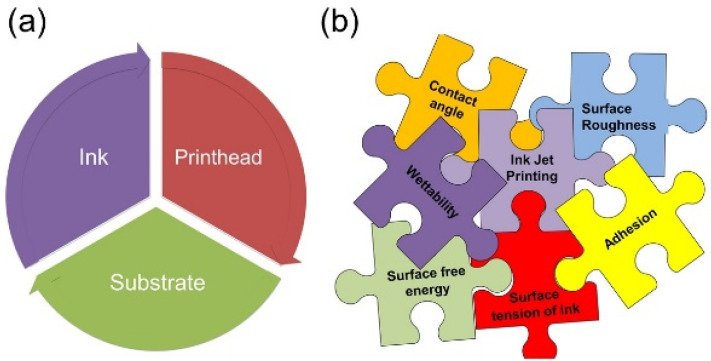
(**a**) The key elements involved in the inkjet printing process. (**b**) The jigsaw to be solved for the understanding of inkjet printing process.

**Figure 2 micromachines-15-00681-f002:**
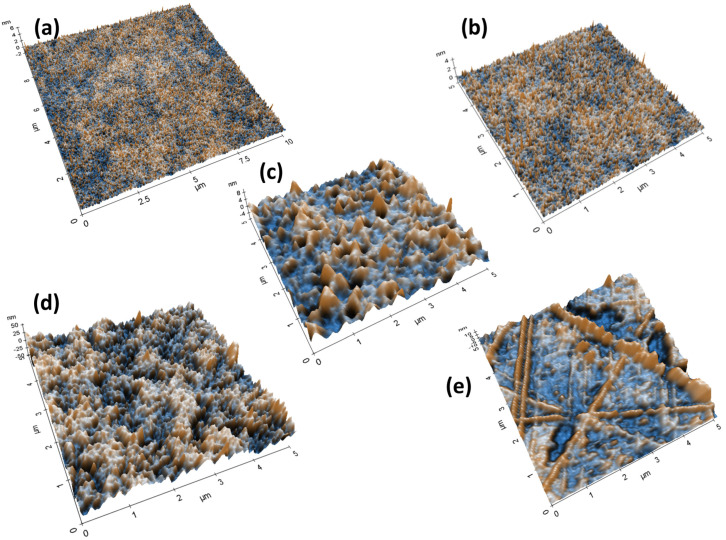
AFM images of (**a**) glass; (**b**) silicon; (**c**) ITO/PET; (**d**) PEL paper; (**e**) cellulose substrates. Surface morphology investigation is conducted over a region of 5 µm × 5 µm area in both x and y directions. X and Y axis units are in µm.

**Figure 3 micromachines-15-00681-f003:**
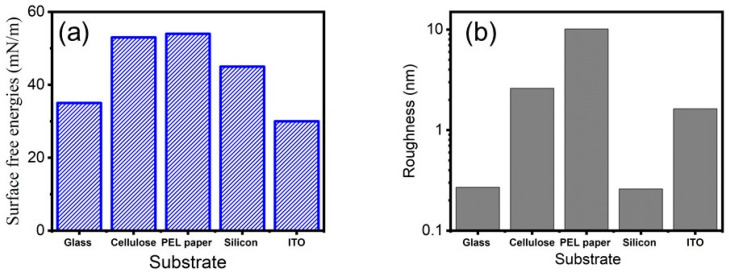
(**a**) Surface-free energy and (**b**) AFM measured roughness of five different types of substrates selected for the investigation in this study.

**Figure 4 micromachines-15-00681-f004:**
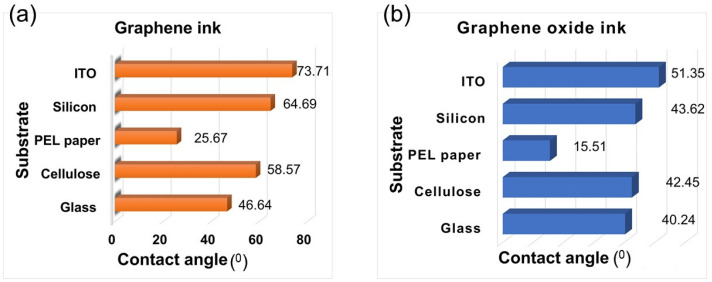
Contact angle measured via sessile drop analysis method for (**a**) graphene and (**b**) graphene oxide inks.

**Figure 5 micromachines-15-00681-f005:**
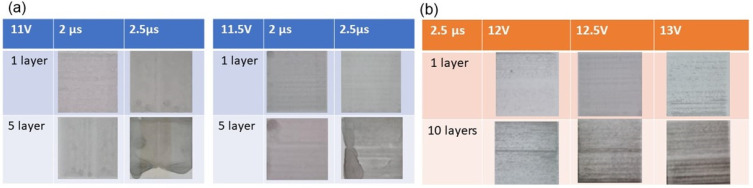
Printing parameters (pulse amplitude and pulse width) were evaluated in order to achieve a viable print for (**a**) graphene ink and (**b**) graphene oxide ink.

**Figure 6 micromachines-15-00681-f006:**
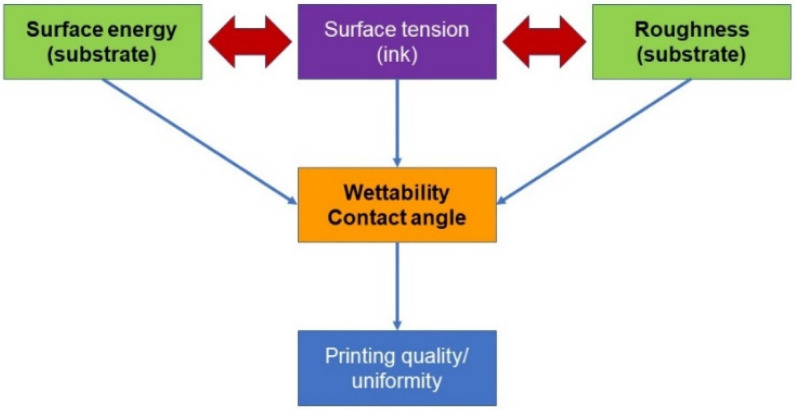
The schematic diagram of the underlying factors behind wettability/printing quality.

**Figure 7 micromachines-15-00681-f007:**
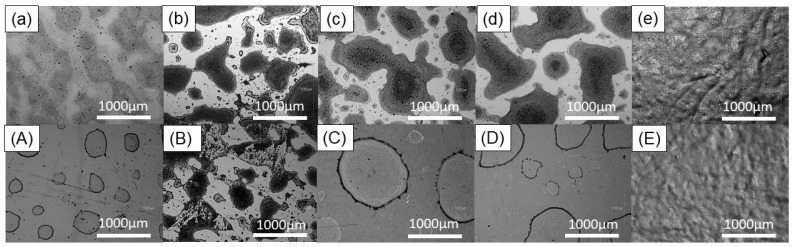
Optical microscope images of five overprinted layers of graphene oxide ink on (**a**) glass; (**b**) silicon; (**c**) cellulose; (**d**) ITO/PET; (**e**) PEL paper, and graphene ink on (**A**) glass; (**B**) silicon; (**C**) cellulose; (**D**) ITO/PET; (**E**) PEL paper.

**Figure 8 micromachines-15-00681-f008:**
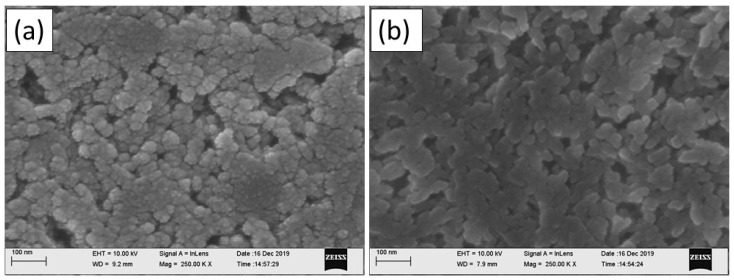
Scanning electron microscope (SEM) images of five printed layers of (**a**) graphene ink and (**b**) graphene oxide ink on PEL paper.

**Table 1 micromachines-15-00681-t001:** Inkjet printing: steps and components.

Step No	Steps in the Printing Process	Component(s) Involved
Step-1	Droplet Formation	Ink’s properties/printhead
Step-2	Droplet Ejection	Printhead/Ink’s properties
Step-3	Deposition of Droplets on substrate and wetting	Ink and Substrate Properties
Step-4	Thin Film Formation	Competition between solvent evaporation and spreading of the ink

**Table 2 micromachines-15-00681-t002:** Measured parameters of both the rGO and G inks.

Parameters	r-Graphene Oxide	Graphene	Required Parameters for IJP
Surface Tension (mN/m)	42	66	30–50
Viscosity (cP) at shear rate 264 s^−1^	14	4	2–20
pH @ measured at 25 °C	7.6	7.4	7

**Table 3 micromachines-15-00681-t003:** Printing parameters evaluated in order to achieve a viable printing.

Voltage Pulse Width	Applied Voltage to the Print Head				
	11 V	11.5 V	12 V	12.5 V	13 V
**2.0 µs**	Graphene ink—faded	Graphene ink—suitable			Graphene oxide ink—faded
**2.5 µs**	Graphene ink—too much ink ejected	Graphene ink—too much ink ejected	Graphene oxide ink—faded	Graphene oxide ink—suitable	Graphene oxide ink—suitable

## Data Availability

The original contributions presented in the study are included in the article, further inquiries can be directed to the corresponding author.

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
