# Peer review of "Influence of Surface Chemical and Topographical Properties on Morphology, Wettability and Surface Coverage of Inkjet-Printed Graphene-Based Materials"

_micromachines, 2024, doi:10.3390/mi15060681_

Round 1

Reviewer 1 Report

Comments and Suggestions for Authors

This article studied the effects of the ink physical properties,the chemical and topographical properties of  5 substrates on the printing qualilty using the water-based graphene and GO inks. Generally the article is well-writen and organized well. The result is analyzed reasonablly. However, as for water-based ink printing, it is better to consider the water-proof of the printing pattern and the adhensive with the substrates. For these goals, authors should discuss the posttreatment as laser-jet printing process to fix these petterns and the interfacial reaction between the ink and the substrates. In addition, there may be some in-situ pre-treatment of the surface of the substrate before printing equiped in the ink-jet printer.  At least, some progress can be reviewed in the introduction. 

Author Response

First of all, we would like to thank you for the provided comments as these allow us to further enhance the quality of our work. The following comments have now been addressed: 

We would like to warmly thank the reviewer for describing our manuscript as “well -written” and “organized well”.

The reviewer is correct that in the main text we did not discuss post-treatment after printing. We intentionally opted not to do so, as we feel that the main scope of the paper is to discuss the influence of both the properties of the substrate and the properties of ink and proving their respective impacts on the quality/surface coverage of graphene and graphene oxide inkjet printed multilayers. However, after validation that the PEL paper is the best substrate (included in this work) a post-treatment at 800C/20 mins was performed for both the graphene and graphene oxide inks with this information being included in another, under review, paper.

Additionally, if we were to introduce the suggested adhesion and in-situ pre-treatment results/discussions, it would significantly expand the main body of text. However, we appreciate your input and will certainly include your suggestion in future research or presentations related to this follow-on work.

We have now amended the introduction according to the reviewer’s comment.

We concur with the reviewer; we have also added the following text:

The substrates selected did not receive any treatment before or after printing.

Reviewer 2 Report

Comments and Suggestions for Authors

In this paper, the impact of both the chemical and topographical properties of substrates and the physical properties of graphene-based inks on the morphology, wettability and surface coverage of the inkjet-printed patterns have been investigated. However, improvements are still needed in the following areas:

1. This paper studied inkjet printing experiments on two rigid substrates and three flexible substrates. However, the reasons for choosing these five substrates are not clearly stated. It is recommended to supplement them completely.

2. The "Results and Discussions" section in this paper measured parameters such as surface tension, viscosity, PH of rGO and G inks. What specific values can these parameters to prove?  It is recommended to supplement them completely.

3. The unit 2µ, 2.5µ in figure 5 of "Results and Discussions" in this paper is incorrect and should be changed to 2µs, 2.5µs.

4. The figure 6 of "Results and Discussions" in this paper explains the relationship between wettability and printing quality, but the explanations are superficial. Can it be further analyzed more deeply? It is recommended to supplement them completely.

5. The figure 7 of "Results and Discussions" in this paper, the comparison of the uniformity of inkjet printing on different substrates is shown. However, it appears that most inkjet printing on substrates exhibit uneven morphology. Perhaps further optimization of control parameters can be used to improve uniformity.

6. The "Results and Discussions" in this paper analyzes the process of inkjet printing, but lacks supporting methods such as formula derivation and simulation experiments. To my knowledge, many teams have conducted relevant research on the liquid rheological process of inkjet printing. It is recommended to refer to relevant papers such as " https://doi.org/10.1007/s10404-022-02621-4" and " DOI10.3390/mi15030333 ".

Comments on the Quality of English Language

The quality of English is suitable for pubilication.

Author Response

Reviewer 2

In this paper, the impact of both the chemical and topographical properties of substrates and the physical properties of graphene-based inks on the morphology, wettability and surface coverage of the inkjet-printed patterns have been investigated. However, improvements are still needed in the following areas:

  1. This paper studied inkjet printing experiments on two rigid substrates and three flexible substrates. However, the reasons for choosing these five substrates are not clearly stated. It is recommended to supplement them completely.

We thank the reviewer for pointing this out. We have now included an explanation to avoid any confusion.

“It is important to highlight that flexible substrates were selected as today they play a crucial role in the field of flexible/paper electronics. On the other hand, silicon and glass are still used in the field of electronics, so in a nutshell this study will provide a comprehensive study covering both types of substrates”.  

  1. The "Results and Discussions" section in this paper measured parameters such as surface tension, viscosity, PH of rGO and G inks. What specific values can these parameters to prove?  It is recommended to supplement them completely.

 Amended accordingly; the required parameters have now been included in Table 2.

  1. The unit 2µ, 2.5µ in figure 5 of "Results and Discussions" in this paper is incorrect and should be changed to 2µs, 2.5µs.

 Amended accordingly.

  1. The figure 6 of "Results and Discussions" in this paper explains the relationship between wettability and printing quality, but the explanations are superficial. Can it be further analyzed more deeply? It is recommended to supplement them completely.

Thank you, the reviewer, for raising this point. We have now added the explanation:

 Another key observation is that the wettability can be defined and characterized by the contact angle [25].  The arriving droplets on the substrate should undergo two steps:  firstly, a liquid-liquid interface evolution and secondly droplets coalescence. Due to the small contact angle these steps are fast and uniform and void free coating printing patterns can be achieved. On the other side, it is more difficult to obtain a high-quality pattern when the contact angle is high. Both the liquid-liquid interface / droplets coalescence are relatively slow and usually needs other factors such as temperature to sustain the coalescence process.

  1. The figure 7 of "Results and Discussions" in this paper, the comparison of the uniformity of inkjet printing on different substrates is shown. However, it appears that most inkjet printing on substrates exhibit uneven morphology. Perhaps further optimization of control parameters can be used to improve uniformity.

The highlighted aspect has been now addressed:

Another key observation is that the roughness of the substrate has a huge impact on the flatness of the printed patterns. It can be clearly seen from figure 7 that both graphene oxide and graphene pattens printed on PEL paper exhibit the most significant uneven topography when compared with the rest of the used substrates.  

  1. The "Results and Discussions" in this paper analyzes the process of inkjet printing, but lacks supporting methods such as formula derivation and simulation experiments. To my knowledge, many teams have conducted relevant research on the liquid rheological process of inkjet printing. It is recommended to refer to relevant papers such as " https://doi.org/10.1007/s10404-022-02621-4" and " DOI10.3390/mi15030333 ".

 We thank the reviewer for pointing this out. We would like to highlight that the main focus of this paper is on experimental results. However, we appreciate your input and will certainly keep your suggestion in mind for future research/work.  

Additionally, there is a significant difference between inkjet printing and electrohydrodynamic printing from the process point of view. In inkjet printing, the cartridge has 300 nozzles with each having a 28 µm diameter with the ink being ejected by the thermal (heating element) principle. On the other side, electrohydrodynamic printing has only a singular nozzle (needle) with a diameter 0.3 to 30 µm with the ink being ejected via applied voltage. In conclusion, it is not relevant to refer to/acknowledge the recommended/suggested papers.

We warmly thank the reviewer for the valuable comments/suggestions.

Reviewer 3 Report

Comments and Suggestions for Authors

This work is about using inkjet printing to print graphene-based inks on various kind of substrates. Although the work doesn't appear to be very novel, yet i do see some merits in this work. However, i feel that the quality of the figures and data presentation could have been done better. The motivation of the work should also be clarified.

Specific comments.

1. It appears that the landscape of printed electronics is not well-discussed. Suggest discussing various type of droplet-jet based printing method such as aerosol jet printing (AJP). Suggest discussing why IJP is selected and more superior compared to other techniques like AJP. You may consider citing the following works for discussion and comparison.

a. S. Agarwala, G. L. Goh, and W. Y. Yeong: 'Aerosol jet printed pH sensor based on carbon nanotubes for flexible electronics', Proceedings of the 3rd International Conference on Progress in Additive Manufacturing (PRO-AM), Nanyang Technological University, Singapore, 2018, 88-94.

b. Jabari, E., & Toyserkani, E. (2015). Micro-scale aerosol-jet printing of graphene interconnects. Carbon91, 321-329.

2. Can the authors add another figure to complement figure 1?

3. In section 2 (methods), suggest providing more information about the test standards that were adopted in this work.

4. What was the sample size used for each test?

5. The formatting of the tables appears to be unprofessionally made. Suggest improving the quality of the tables.

6. Figure 2, the text next the axes is too small.

7. Following query 4, it appears that there are missing error bars in many of the plots.

8. Figure 4, suggest using the symbol for degree instead.

9. Table 3, why is the table color-coded without a legend to explain the color?

10. Figure 5, i do not know what i should be looking out for. suggest enlarging the figure and place arrows to highlight what to look out for?

11. Figure 6 is a figure that can be found in other papers, suggest citing appropriately if it is adopted from any other sources.

12. Missing scale bars for figure 7.

13. Suggest evaluating the adhesion of the printed films on each substrate.

14. It is not clear why graphene-based inks were chosen in this work?

15. Also, there are existing reviews that discussed the design considerations for printed electronics. Suggest discussing and citing them ( https://doi.org/10.1002/aelm.202100445, DOI 10.1088/2058-8585/abc8ca) to enrich the discussion in the introduction.

Comments on the Quality of English Language

nil

Author Response

Reviewer 3:

This work is about using inkjet printing to print graphene-based inks on various kind of substrates. Although the work doesn't appear to be very novel, yet i do see some merits in this work. However, i feel that the quality of the figures and data presentation could have been done better. The motivation of the work should also be clarified.

Specific comments.

  1. It appears that the landscape of printed electronics is not well-discussed.

This has been now addressed:

“…and nowadays is intensively explored for the deposition of functional materials and electronic devices with a huge potential to fully or partially replace traditional manufacturing technologies. The impact of 3D printer technologies in the landscape of electronics has been presented in the comprehensive review papers [2,3,4]”

  1. Suggest discussing various type of droplet-jet based printing method such as aerosol jet printing (AJP). Suggest discussing why IJP is selected and more superior compared to other techniques like AJP. You may consider citing the following works for discussion and comparison.
  2. S. Agarwala, G. L. Goh, and W. Y. Yeong: 'Aerosol jet printed pH sensor based on carbon nanotubes for flexible electronics', Proceedings of the 3rd International Conference on Progress in Additive Manufacturing (PRO-AM), Nanyang Technological University, Singapore, 2018, 88-94.
  3. Jabari, E., & Toyserkani, E. (2015). Micro-scale aerosol-jet printing of graphene interconnects. Carbon91, 321-329.

We thank the reviewer for pointing this out. The main benefits of inkjet printing are already highlighted in the manuscript.

Printed electronics are emerging as a novel and sustainable platform for the fabrication of electronic devices, and Inkjet Printing (IJP) fits perfectly with this framework. The main benefits of inkjet printing manufacturing include: reduction of wasted material, cost effectiveness, compatibility with a wide range of substrates, digital and additive deposition, mask-less depositions, as well as small to large area deposition.  [1]

Additionally, the first author (IS) has 10 years of experience with inkjet printing alongside a long-term collaboration with Printed Electronics Ltd and this collaboration has been reflected on the recent grant award (The Royal Society SIF\R2\2320002 - Royal Society Short Industry Fellowships).

  1. Can the authors add another figure to complement figure 1?

After careful consideration, we regret to inform you that we are unable to incorporate an additional figure as we think that figure 1 currently is relevant to the work. However, a graphical abstract will be included.

  1. In section 2 (methods), suggest providing more information about the test standards that were adopted in this work.

Additionally, the viscometer, pH meter, torsion balance and the optical tensiometer were calibrated before the tests were performed.

  1. What was the sample size used for each test?

Now amended:

The viscosity was evaluated by the Brookfield DV2T viscometer, using a small sample adapter with a sample volume of 16 ml and SSA 18 spindle. The surface tension of the inks was measured using a tensiometer (Torsion Balance), 30 ml of ink was examined with a platinum ring. Yet, the pH of 30 ml ink was tested with a Jenway 3520-pH meter.

  1. The formatting of the tables appears to be unprofessionally made. Suggest improving the quality of the tables.

The quality of the tables has been improved.

  1. Figure 2, the text next the axes is too small.

The images presented here are acquired directly from the AFM software and are uneditable. However, we have included the following descriptive text in the figure caption: “Surface morphology investigation is conducted over a region of 5µm × 5µm area in both x and y directions. X and y axis units are in µm”

  1. Following query 4, it appears that there are missing error bars in many of the plots.

In this study, our focus was primarily on exploring the fundamental concepts of underlying science rather than on precise quantitative analysis. However, we appreciate your suggestion and plan to incorporate detailed error analysis, including error bars, in our future work dedicated to substrate variability and repeatability.

Your feedback is invaluable, and we are committed to addressing this aspect in our follow-up research and subsequent publication.

  1. Figure 4, suggest using the symbol for degree instead.

Amended.

  1. Table 3, why is the table color-coded without a legend to explain the color?

To eliminate any confusion the table has been modified.

  1. Figure 5, i do not know what i should be looking out for. suggest enlarging the figure and place arrows to highlight what to look out for?

These pictures were included to support (via optical microscope images) the evaluation included in table 3 and hence illustrate the impact of printing parameters i.e pulse amplitude and pulse width on the quality of the printing patterns. There is not a particular point to be highlighted here.

  1. Figure 6 is a figure that can be found in other papers, suggest citing appropriately if it is adopted from any other sources.

This diagram was made by us and we are not aware that this figure can be found in other papers, can you please provide the reference?   

  1. Missing scale bars for figure 7.

Amended accordingly.

  1. Suggest evaluating the adhesion of the printed films on each substrate.

The adhesion of both the inks on selected substrates were performed but we didn’t include these results in this manuscript as these are not in the scope of this work. The obtained adhesion results are part of another research paper that will be submitted soon. 

  1. It is not clear why graphene-based inks were chosen in this work?

In a nutshell, the inkjet printing of graphene-based inks has a huge potential for the development of unique and practical electronic components.

  1. Also, there are existing reviews that discussed the design considerations for printed electronics. Suggest discussing and citing them (https://doi.org/10.1002/aelm.202100445, DOI10.1088/2058-8585/abc8ca) to enrich the discussion in the introduction.

The suggested papers were included in the introduction of references 2 and 3.

We warmly thank the reviewer for the valuable comments/suggestions.

Round 2

Reviewer 3 Report

Comments and Suggestions for Authors

The authors have addressed the queries satisfactorily. The manuscript can therefore be recommended for acceptance for publication.

Comments on the Quality of English Language

nil